# Parallelogram Excision: An Algorithmic Approach for Excision Designs in High-Tension Skin Areas

**DOI:** 10.3390/healthcare11192624

**Published:** 2023-09-26

**Authors:** Francesco Costa, Filippo Boriani, Syed Haroon Ali Shah, Jeyaram Srinivasan

**Affiliations:** 1BST Biomedical Science and Technologies Lab, IRCCS Istituto Ortopedico Rizzoli, 40136 Bologna, Italy; francesco.costa@gmail.com; 2Department of Plastic Surgery and Microsurgery, University Hospital of Cagliari, University of Cagliari, 09124 Cagliari, Italy; 3Department of Plastic Surgery, Royal Preston Hospital, Preston PR2 9HT, UK; syedharoonalishah@gmail.com (S.H.A.S.); jsrinivasan@doctors.org.uk (J.S.)

**Keywords:** skin excision, plastic surgery, parallelogram excision, sarcoma, orthopedics

## Abstract

Introduction: The excision of lesions that are not oriented along the skin tension lines may cause the surgeon to design extremely broad elliptical preoperative markings, with the intent to follow the tension lines as recommended for the best postoperative course and the best quality scars. The aim of this study is to describe and clinically apply a new surgical technique called the parallelogram excision technique, in which the traditional ellipse with a major axis parallel to the tension lines is converted into a parallelogram whose lesser sides are coincident with the local skin tension lines. This technique was specifically conceived for lesions whose major axis is non-coincident with skin tension lines, and the primary advantage is that it reduces the amount of healthy tissue excised. Methods: Preliminarily to this clinical study, a comparative geometrical analysis was conducted between various excision shapes and angles using Geometry Pad version 2.7.10 (Bytes Arithmetic LLC) and verifying the data obtained through AutoCAD 2D 2016 (Autodesk, San Rafael, CA, USA), with the purpose of optimizing the technique from a geometrical point of view. A comparison was performed between the theoretical traditional elliptical excision and the hypothetical parallelogram excision. A pilot proof of concept clinical study was performed to verify the validity of the excisional design proposed. The patients considered for parallelogram excision suffered from skin lesions with a diameter no greater than 4 cm and oriented 45° to 60° with respect to tension lines. In order to limit variability, patients’ ages were between 40 and 80, and the selected areas were limbs, sternum and dorsum. Scar quality was assessed with the validated POSAS method at 6 months post-operation. Results: The geometrical analysis of the parallelogram’s design showed that it allows a diminution of the excised healthy skin compared to the traditional ellipse. The clinical series included 16 patients, with a mean age of 63.5. Of these, nine patients were men and seven were women. Diagnoses included basal cell carcinoma in seven cases, dysplastic naevus in five patients, Bowen’s disease in three individuals, and one case where a wider excision of a malignant melanoma was performed. Six-month follow up results showed: (1) an uneventful postoperative course; (2) good scar healing with an observer’s POSAS median score of 16 and a patient’s POSAS median score of 19; and (3) complete excision of lesions. Conclusions: When indicated, the parallelogram excision technique appears to be a good option for the excision and primary closure of skin lesions that are not parallel to skin tension lines, since it allows a reproducible and surgeon-friendly method of preoperative marking and implies a favorable use of the local tension, which determines good quality scars. The amount of healthy tissue removed is smaller compared to traditional elliptic excisions.

## 1. Introduction

Plastic surgeons are perhaps the “surgeons of wound closure”, since they possess the necessary skills and expertise necessary to achieve this goal in challenging clinical situations, particularly when dealing with major defects or complex areas in terms of local skin tension. The traditional reconstructive ladder is an essential concept and nowadays still informs plastic surgeons on how to approach the choice of wound closure [1]. It has been developed through research and experience, thereby resulting in a scheme that recommends increasingly complex methods of wound closure and reconstruction. The aim is to achieve wound healing and restore function with the smallest amount of morbidity [1].

Recent technological advancements have allowed the reconstructive ladder to evolve [2,3] while still holding true to its primary principle. For example, the use of topical negative pressure has allowed the addition of a new rung of wound closure—albeit as a temporizing measure rather than as a ‘technique’. The concept of the ladder continues to develop new interpretations, such as the reconstructive ‘elevator’, which refers to a surgeon’s flexibility in the selection of the best reconstructive option to choose from, rather than the step-wise approach of exhausting all basic principles of closure first [2,3]. An even more recent evolution of the ladder–elevator metaphor proposed by Knobloch is the reconstructive clockwork [4], which describes the novel and expectable progress and expansion in the armamentarium of the plastic surgeons in terms of regenerative techniques and vascularized composite allotransplantation.

The external appearance of the human body is outlined by the underlying bony skeletal scaffold, which is covered by skin. To adapt to this complex morphology, the cutaneous covering has to be both elastic and viscous, in order to allow deformation and return to its previous form. From a mechanical point of view, it has to be both flexible and strong.

The vectors of tension throughout the skin are specifically related to the movement and volume of the underlying shape, and high cutaneous tension is most frequently associated with imperfect or pathologic scar formation post-surgery or following lacerations.

Unfortunately, the quantitative determination of skin tension is neither practical nor reliable in clinical situations and settings. On the other hand, skin wrinkles or specific lines have been commonly employed as a surrogate superficial map of tension vectors. A variety of skin lines have been considered over time [5], whilst relaxed skin tension lines (RSTL) and Langer’s lines are the most commonly taken into consideration. Interrupting the cutaneous surface with a punch excision usually determines a circular wound that immediately evolves into an elliptical defect. If a multitude of punch excisions are carried out and the major axes of these ellipses are associated in continuity, the resulting line is deemed to be a Langer’s line [6,7]. These vectors run parallel to the main collagen fascicles of the dermis and sometimes, but not always and not necessarily, follow natural cutaneous creases.

A relaxed skin tension line (RSTL) is a groove typically resulting from the skin being pinched and relaxed in the absence of local tension [6,7].

Physiologically and anatomically, the cutaneous mantle appears to be most extensible perpendicular to RSTLs, and this suggests that the tension is minimal in cases where the excision is performed along RSTLs.

RSTLs and Langer’s lines run parallel across many regions of the body, whilst they are significantly noncoincident in mechanically sophisticated areas such as the temple, the lateral canthal region and the mouth corner. The term used to describe this feature of the human cutaneous map is anisotropy, which refers to the variability of skin tension according to the various body sites. The reason behind this is a biological and biomechanical feature, and is related to the amount, the quality and the direction of collagen and elastin fibers in each body area, with variations depending on gender and age [8,9].

An appropriate masterplan of cutaneous vectors of tension is essential for correct incision, and even more so for excision planning and outlines.

Inappropriate surgical planning and markings are the most common and main reason for pathologic scarring (scar hypertrophy and keloid) occurring in the course of wound remodeling. Excessive tension across a healing wound/maturing scar may result in pulling the margins apart.

As a pathophysiological consequence, the wound acquires a tendency to hold itself together more tensely. Macroscopically, the scar tends to become hypertrophic. Microscopically, this turns out to be an increased collagen deposition.

The critical aspect is in the timing. Since hypertrophic responses are noticed months after an incision has been performed, the surgeon cannot revert and correct the errors made.

In specific areas of high cutaneous tension, a hypertrophic reaction is unavoidable, irrespective of the direction of incision (i.e., sternal area and shoulders).

Similarly, and on a larger ground, incisions should almost never involve the extensor aspects of the joints. The opposite holds true; flexor grooves are elect directions for surgical incisions.

Many body regions represent challenging areas, with a high incidence of skin and soft tissue tumors, and high local tension causing a risk of wound dehiscence or pathologic wound healing due to the retracting effect of scars. Limbs, dorsum and sternal area are body regions with such features.

The fact that the excision ellipse of a skin lesion needs to be outlined with its major diagonal following the major axis of the neoplasm is an established principle in plastic surgery. The recent studies by Paul [10] on biodynamic excisional skin tension (BEST) lines have determined with a scientific method the mapping of the tension lines of the body, untangling a topic which was treated more as a narrative and anecdotal story rather than evidence-based knowledge as it is now. This means that when performing excisions, the usage of BEST lines is becoming increasingly recommended.

The problem occurs when the major axis of the lesion is not parallel to the Langer’s lines, RSTLs or BEST lines, as the excisional ellipse might be difficult or impossible to close and the deriving scar might be at risk of the aforementioned complications.

In some of these cases, the traditional options offered by plastic surgery are represented by skin grafts or flaps. When the major axis of the lesion and the tension lines do not correspond, most of the time surgeons are embarrassed to be left with the choice of an excision design that does not respect the tension lines or performing an excision according to these lines at the price of a far broader defect. In particular, trainees or junior surgeons, with less clinical experience in their background, would benefit from an algorithmic approach on how to deal with these situations. For this purpose, the authors have studied and clinically applied a model of excisional design which can guide the surgeon when the major axis of the lesion would require an excessively wide ellipse, or an excision non-compliant with the local map of tension lines.

This proof of concept could be proposed as a new reconstructive rung to the ladder or as a development of the basic rung “direct closure”. The innovation described has been defined as the ‘parallelogram’ excision, as illustrated by Figure 1, and consists of a method that allows surgeons to optimize the elliptical excision design of a lesion whose major axis is not parallel to the RSTL.

## 2. Materials and Methods

Preliminarily to this clinical study, a comparative geometrical analysis was conducted between various excision shapes and angles using Geometry Pad version 2.7.10 (Bytes Arithmetic LLC), and the data obtained were verified using AutoCAD 2D 2016 (Autodesk, San Rafael, CA, USA), with the intent to optimize the technique from a geometrical point of view. A comparison was performed between the theoretical traditional elliptical excision and the hypothetical parallelogram excision. The parameters evaluated were total excision area, theoretical length of the final wound, margin distance and healthy excised area. These parameters were compared for the two scenarios (traditional ellipse versus parallelogram) in a variety of angles between the ellipse major axis and the direction of the RSTL.

For this theoretical purpose, we have defined the lesion as an ellipse that has its major axis twice as long as the minor and forming with the RSTL an intersection angle (orientation angle) variable between 30 and 90 degrees.

After the preclinical step, all patients consecutively accessing the authors’ departments of plastic surgery between January 2022 and March 2023 due to skin or soft tissue neoplasms of the limbs, dorsum or sternum were considered for inclusion in this study.

The reason for choosing these body areas was to limit the factors related to anatomic variability of skin tension and elasticity. In particular, the face was excluded because it is a very peculiar area, with critical structures and a lower threshold for upgrading the reconstruction to local flaps or grafts, as even minimal tension may distort the anatomy (e.g., areas surrounding the nose, the eyes and the mouth). The abdomen has no bones underneath, which makes the local elasticity much higher. The chest has the nipple–areola complex, which has the same issues as the facial area.

Inclusion criteria were a lesion whose major diameter was equal to or smaller than 4 cm; the orientation of the major axis was angulated between 45 and 60 degrees compared to the local relaxed skin tension lines (RSTL); and patient’s age was between 40 and 80. These age limits were chosen in order to limit the variability of skin elasticity based on the patient’s age, which is a scientifically proven factor determining variations in cutaneous elasticity and tension, as described by Ayadh et al. [9]. It was also considered that the age window defined has the highest incidence of skin tumors.

Exclusion criteria were recurrent lesions and patients with anelastic skin, as evaluated through a pinch test. This study was conducted according to the Helsinki Declaration and no Institutional Revisory Board (IRB) approval was necessary, as per local policies on clinical studies. This was because this study was granted IRB exemption due to the type of risks run by participants, which was quantified as minimal/null (this study aimed at creating an algorithmic guideline for improving the process of drawing the excision lines in patients who would have undergone the surgical procedure anyway, and therefore no innovative treatment was administered nor additional risk caused to patients).

All cases were evaluated in terms of wound healing complications. The scar quality was evaluated at 9 months with the POSAS (Patient Observer Scar Assessment Scale) [11] method for scar evaluation by observer and patient.

This scale has a 6 item patient component and a 6 item observer component (each with a score of 6 indicating the best and 60 indicating the worst scar imaginable).

The observer’s part of the scale was assigned to a plastic surgeon who was not involved in the surgical procedure of the patient.

### Surgical Markings and Technique

Pre-operative parallelogram-shape marking of the patient was designed around the lesion in each case to excise the minimum area of tissue and to achieve direct closure as parallel as possible to relaxed skin tension lines.

The preoperative marking of the parallelogram excision, shown in Figure 2a–c, has to be conducted as follows:Identify the direction of the major axis of the ellipse circumscribing the lesion (plus excision margins) and draw two tangents parallel to that direction (Figure 2a);Draw two other tangents to the ellipse parallel to the RSTL (Figure 2b);Intersect these 4 lines finding 4 vertices and outline the parallelogram (Figure 2c).

All excisions were carried out under local anesthesia as a day case procedure.

**Figure 2 healthcare-11-02624-f002:**
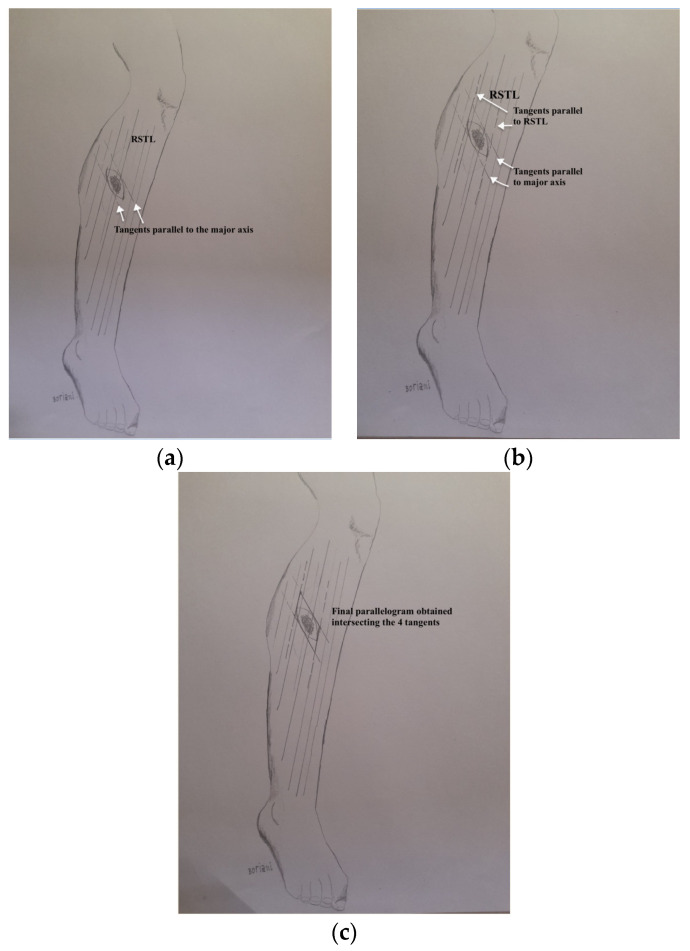
(**a**–**c**) Step-by-step preoperative markings for a parallelogram excision in a leg with major axis oriented 50° to RSTL. After definition of lesion borders and correct oncological margins, tangents parallel to the major axis of the excision are drawn (**a**). Tangents parallel to the RSTL are outlined (**b**). The parallelogram excision design derives from the intersection of the four tangents (**c**).

## 3. Results

The software analysis took into account a number of hypothetical excision orientations, as illustrated in Table 1, which shows that in a variety of hypothetical excisions that are not parallel to the skin tension lines, the parallelogram technique requires less removal of healthy skin.

With regard to the clinical series, in the selected period, nineteen patients were considered for inclusion, but three were excluded due to recurrence (two) and anelastic skin (one) and therefore underwent reconstruction through skin graft.

Demographics of the included 16 patients is shown in Table 2, whilst orientation, site and size of the excisions are illustrated in Table 3.

All included patients had an uneventful postoperative course and all scars healed with no complications, including dehiscence, hypertrophy, hypotrophy or keloid. Out of the included sixteen cases, seven were basal cell carcinomas, five were dysplastic nevi, three were Bowen’s disease, and one was a wider excision for malignant melanoma. The oncological lateral margins taken were 3 mms for the nevi and 4 to 10 mms in the other cases. All specimens were completely excised.

The POSAS scale was used to evaluate all cases except the most recent one on the dorsum, which has not reached the 6 months follow up yet, and the results are illustrated in Table 4.

Figure 3a–c shows the preoperative markings of a wider excision of malignant melanoma in a 67 years old man. Due to the excision orientation of 45° degrees with respect to BEST lines, a parallelogram was obtained from the intersection between tangents to the ellipse parallel to major axis and BEST lines. Figure 4a–c show the parallelogram excision wound, the approximation of margins and the immediate postoperative result. A video shows the vectors by which the margins were approximated in order to avoid standing cones. Figure 5 illustrates the result 3 weeks postoperatively.

## 4. Discussion

The aim of cutaneous oncologic surgery is the complete excision of neoplasms and the reconstruction of resultant defects, to let patients develop a healed scar and allow them to restart pre-surgery life in terms of work and other activities [12]. In cases of minor defects, primary closure is possible; however, when defects are too broad to be repaired primarily, the various options offered by the reconstructive ladder are feasible, including full and partial thickness skin grafts [13,14], halo grafts [15], local flaps, regional flaps, perforator-based island flaps such as the Keystone Flap [16] and opposed multilobed flaps [17]. For excisional wounds that cannot be sutured primarily, in cases where the choice of a skin graft or a local flap is not feasible, microvascular free tissue transfer [18] is generally the indicated solution to be utilized.

Even though the repair of defects with the least tension is well-proven as determining the best results in terms of both wound healing and scar-related processes, the excisional vectors that minimize tension have often been controversial and far from a general consensus in the scientific community. It has been noted that a number of books define and outline cutaneous tension lines, such as Langer’s lines, diversely, and lines currently used for surgical excision may be based on a poor scientific ground [19].

Borges [7] initially reported the idea of utilizing RSTLs for the removal of skin lesions on the head and neck region. RSTLs rely on the concept that with the skin relaxed, grooves appear. Moreover, these vectors become more visible by pinching skin and detecting the direction of the ridges and creases [7].

Nevertheless, on the rest of the body or areas such as the shoulders, sternum and lower limbs, Borges [8] usually relied on Kraissl’s lines [20], which basically consist of wrinkle directions.

Another important concept on the topic derives from recent studies on wound tension in surgical wounds, in which a hypothesis was described that excisional and incisional lines need to be identified and approached differently [19,21].

Wrinkles and creases that are adequate for surgical incisions may not be sufficiently tension-bearing to let them deal with the increased load posed by major excisional surgery [22].

This is partially in contrast to the original study on the matter by Langer [23] in which no mention was made about a possible difference between excision and incision, since the study methodology was only based on incision tests.

Even though the skin is anatomically viewed as a unique organ and as our largest organ, a biomaterial engineering perspective reconsiders our integuments as a composite tissue with anisotropy and elastic behavior only at low-tension levels.

Body regions such as the feet and lower limbs, due to major weight-bearing physiology, show a cutaneous mantle that reveals greater viscoelastic behavior, i.e., tension depends on patient’s age and load [22]. In lower limbs with sun damage, not only can modifications due to age and UV radiations be noticed, but also the underlying elastin system demonstrates a certain degree of deterioration over time [24].

In the lower limb region, especially in the legs, distally to the knees, the effect of improper orientation of excision markings is relevant, as it may involve the choice between primary closure and a skin graft, which necessitates a longer healing time and suboptimal aesthetic outcome. Hence, the correct orientation of the excisional ellipse when planning oncologic skin surgery is of paramount importance.

Scientific research exploring wound strain by means of tensiometer on different body sites has already proven that for exclusively incisional wounds or excisions with a diameter smaller than 7–8 mm, there is very little difference in wound-closing tension, irrespective of the direction of the ellipse major axis [25].

Skin has often been qualified as anisotropic, but in fact, in areas adjacent to the bone such as the pretibial region, the cutaneous mantle does show orthotropy, i.e., some symmetry (in terms of tension of the excisional ellipse) relative to two perpendicular planes, which depends on the main and preferential orientation of collagen fibers [26].

Tensiometric analyses in other areas like the scalp revealed an evident directional predilection relative to decreased closing tension.

As noticed by Borges [7], Langer’s lines were originally studied and defined [20] in cadavers that had undergone rigor mortis and this bias was not taken into consideration; therefore, they can hardly be defined lines of relaxation.

In the lower limb, Borges [7] used wrinkles as indicators of lines with the least tension. In anatomical sites other than the face, Borges’s lines indeed respect Langer’s lines, and in directions noncompliant with Langer’s lines, they are hindered by the cutaneous tension that results in rendering them irregular [27].

In the controversial lower limb region, Paul [28] demonstrated that biodynamic excisional skin tension (BEST) lines run in the vertical direction.

The parallelogram technique is a surgical procedure designed to allow excision and primary closure of skin defects sometimes deemed in need of skin grafting or a loco-regional flap due to the extension of the excision area and/or tightness of the skin, in cases where the orientation of the tension lines is not coincident with the major axis of the lesion. The geometrical characteristics of the total excision area are improved by accepting a slight rotation of the wound margins toward tension lines. The purpose of parallelogram excision is to minimize the tension between wound edges and extend the direct closure applicability, avoiding the use of further steps of the reconstructive ladder. This clinical study has shown that the proposed technique attains good quality scars (POSAS scores are close to their lowest possible scores, which indicates the best possible scar), with a negligible risk of complications.

There are limitations to this study: the case series is small, the follow up is short and there is no control group with which to compare the parallelogram technique. Future studies should address these issues by performing a comparative analysis of a larger series of parallelogram excisions and traditional broader ellipses, to further confirm if the technique proposed is advantageous in terms of scar length and quality, postoperative course and complications.

This method is particularly valid for surgeons in their initial approach to skin excisions and junior doctors who tend to literally and mechanically apply the concept of orientating the ellipse according to the RSTL. For these colleagues, an algorithmic approach to preoperative markings is desirable. For surgeons with more surgical experience, technical background, and empirical knowledge and expertise, it has to be acknowledged that their ellipse is not always drawn with the major axis exactly parallel to the RSTL, as they master the pinch test and the ability to make designs that allow direct closure, save excisional tissue and simultaneously avoid standing cones.

The applicability of this technique depends on the initial direction of the skin lesion in relation to the RSTL, and this aspect strongly affected the selection of the patients in this present study.

Based on geometrical considerations, the possible advantages of using the parallelogram technique can be summarized:Reduction of the healthy area that is sacrificed;Reduction of the distance between edges and spreading of tension in two equidistant points;Reduction of the theoretical length of the scar, with increasing benefit going from 45° to 90° of the orientation angle between the lesion and the RSTL;Slight rotation of the initial direction of the lesion towards the RSTL.

During the surgical series we clinically observed:Ability to perform direct wound closure that may otherwise need a higher step in the reconstruction ladder such as skin graft or a flap;Orientation of the resultant scar toward RSTL;Reduction in the operating time thanks to reproducible preoperative markings and possible avoidance of skin grafts and flaps.

However, the extent of the advantages discussed above is dependent on the orientation angle of the lesion to the RSTL. In fact, similar to the Z-plasty [29], in which angles are crucial in defining the benefits of the technique, our parallelogram excision is more suitable when applied to orientation angles within 45 and 60 degrees.

Geometrically speaking, the overall benefits of using the parallelogram increase as the orientation angle goes from 40 to 90 degrees, but from 60 to 90 degrees these benefits are limited by the behaviour of the skin, due to the high risk of dog ears, as pointed out in the publication by Sharma [30], which comments on a similar issue in relation to Z-plasty, and arrives at the same conclusion.

For these reasons we strongly suggest using the parallelogram technique in lesions with the orientation angle between 45 and 60 degrees, which have the best balance between geometrical benefits and the low risk of dog ears. This is a disadvantage of the parallelogram technique, but future studies will evaluate if and how the applicability can be extended to other orientations and to other age ranges, as the patients included in this present study were only between 40 and 80.

In conclusion, when indicated, the parallelogram excision technique appears to be a good option for the excision and primary closure of cutaneous defects that are not parallel to skin tension lines, since it allows a reproducible and surgeon-friendly method of preoperative marking and involves a favorable use of the local tension, which determines good quality scars. The amount of healthy tissue removed is less compared to traditional elliptic excisions. All patients who underwent this excision method had excellently healed final scars and were satisfied with the results.

As per current evidence, the target population should be between 40 and 80 years old, with a lesion not exceeding 4 cm of major diameter, and orientation with respect to tension lines between 45° and 60°. Furthermore, in selected cases, the parallelogram option may avoid skin grafts or loco-regional flaps and donor site morbidity, thereby reducing operative time.

## Figures and Tables

**Figure 1 healthcare-11-02624-f001:**
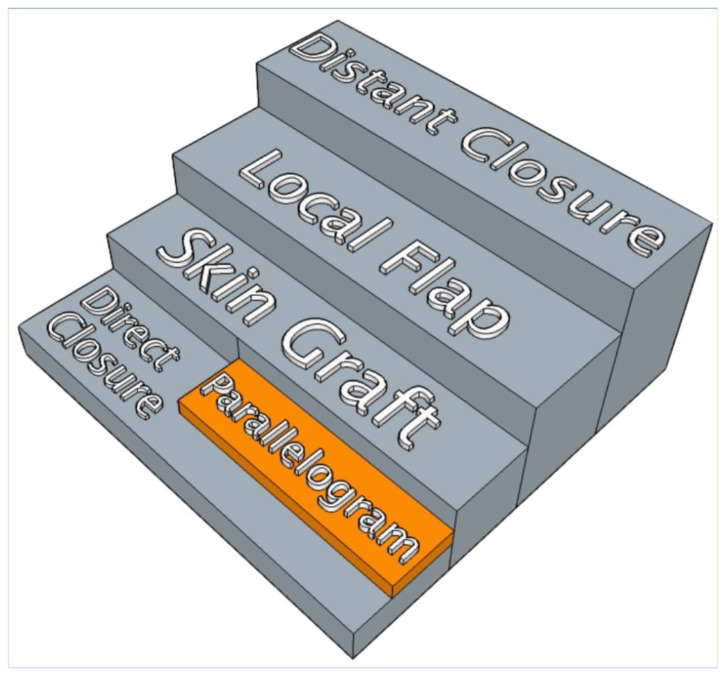
Reconstructive ladder with the proposed new rung.

**Figure 3 healthcare-11-02624-f003:**
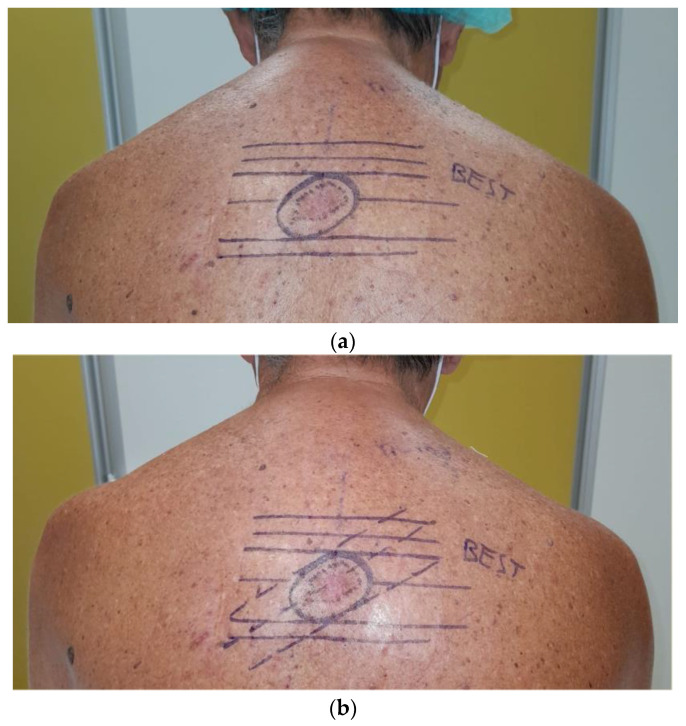
(**a**–**c**) Preoperative markings of a wider excision of malignant melanoma in a 67 years old man. Due to excision orientation of 45° degrees with respect to BEST lines, a parallelogram was obtained from intersection between tangents to the ellipse, parallel to its major axis and BEST lines.

**Figure 4 healthcare-11-02624-f004:**
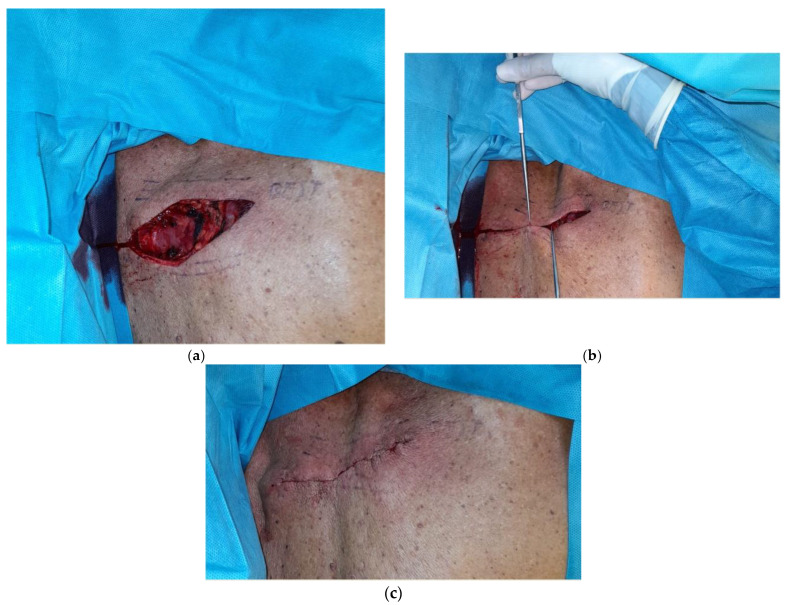
(**a**–**c**) Parallelogram excision wound (**a**), the approximation of margins (**b**) and the immediate postoperative result (**c**). **Video:** vectors by which the margins are approximated in order to avoid standing cones.

**Figure 5 healthcare-11-02624-f005:**
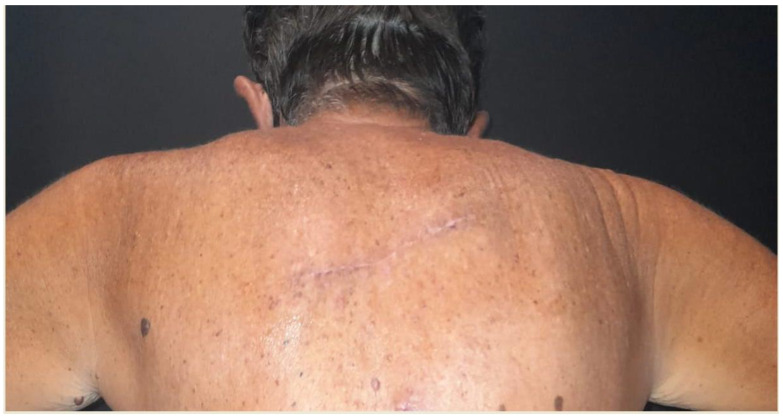
Result 3 weeks postoperatively.

**Table 1 healthcare-11-02624-t001:** Software-based simulations of a variety of excisions with diverse orientations as compared to skin tension lines and various dimensions. Measurements are expressed in cm (length) and cm^2^ (surface). In most cases, the parallelogram configuration allows for better preservation of tissue and a smaller excision.

Orientation Angle	Considered Parameters	Elliptic Excision	Parallelogram Excision	Gain in Local Tissue Preservation with the Parallelogram
30°	Total Excision Area	71.2	42.5	28%
	Theoretical Length	16.5	18	−9%
	Margin Distance	5.5	4.2	25%
	Healthy Excised Area	46	17	62%
45°	Total Excision Area	97.8	35.9	60%
	Theoretical Length	19.3	13.6	30%
	Margin Distance	6.4	4.4	32%
	Healthy Excised Area	73	11	85%
60°	Total Excision Area	125.5	33.5	89%
	Theoretical Length	21.9	11.4	48%
	Margin Distance	7.3	4.5	39%
	Healthy Excised Area	151	59	61%
75°	Total Excision Area	148	32.6	112%
	Theoretical Length	24	10	58%
	Margin Distance	7.9	5.1	35%
	Healthy Excised Area	123	7	94%
90°	Total Excision Area	150.8	44.8	103%
	Theoretical Length	24	8.9	63%
	Margin Distance	8	5.7	29%
	Healthy Excised Area	126	20	84%

**Table 2 healthcare-11-02624-t002:** Demographics of the included patients. IQR: interquartile range. MM: malignant melanoma.

Age median (min to max), IQR	63.5 (45–79), 13
Gender	9M 7F
BMI in kg/m^2^ median (min to max), IQR	27.5 (23– 34), 6.25
Smoking N and %	1 6
Comorbidities N and %	
Hypertension	10 63
Atrial fibrillation	5 31
Hypothyroidism	2 13
Diabetes type 2	2 13
Diagnosis N and %	
Basal Cell Carcinoma	7 44
Dysplastic naevus	5 31
Bowen’s disease	3 19
Wider excision for malignant melanoma for MM	1 6

**Table 3 healthcare-11-02624-t003:** Size, site and orientation of excisions included in the series.

Case	Size of Parallelogram (Major by Minor Axis in cm)	Site	Angle with RSTL
1	9 × 5	Upper arm	60°
2	10 × 5	Upper arm	50°
3	8 × 4	Upper arm	45°
4	10 × 6	Forearm	60°
5	8 × 5	Forearm	70°
6	10 × 7	Thigh	65°
7	8 × 6	Thigh	70°
8	7 × 3	Thigh	50°
9	8 × 4	Leg	45°
10	9 × 6	Leg	60°
11	10 × 5	Leg	40°
12	8 × 5	Leg	70°
13	9 × 5	Sternum	60°
14	7 × 4	Sternum	50°
15	6 × 4	Sternum	60°
16	15 × 7	Dorsum	45°

**Table 4 healthcare-11-02624-t004:** Scar evaluation through Observer Scar Assessment Scale (OSAS) and Patient Scar Assessment Scale (PSAS) at 6 months postoperation, showing good quality scars according to both evaluations.

	Median	Min to Max	IQR
OSAS	16	12 to 18	1.5
PSAS	19	14 to 24	3.5

## Data Availability

Data available on request due to restrictions related to patients’ privacy.

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
