# Peer review of "Parallelogram Excision: An Algorithmic Approach for Excision Designs in High-Tension Skin Areas"

_healthcare, 2023, doi:10.3390/healthcare11192624_

Round 1

Reviewer 1 Report

The authors described a technique, called “parallelogram excision technique” to reconstruct the skin defects from tumor excision. They used geometric measurement to demonstrate the improvement of the total excision area and final scar length which has good clinical values. However, there are some more flaws should be clarified or revised as follows:

1.      The figures were too small and couldn’t be readable on the manuscript. Please re-upload these figures with better resolution to clearly demonstrate your ideas.

2.      The study included 15 cases but none of the clinical pictures were shown. I strongly recommend the authors to present some cases with the excision design and outcome, and I believe this will make the study more convincing.

3.      The lesion size, location and the angles of the parallelogram excision of the clinical cases should be presented. A table to summarize the information can be considered.

In the discussion, the authors stated “Geometrically speaking, the overall benefits of using parallelogram increase as the orientation angle goes from 40 to 90 degrees, but from 60 to 90 degrees these benefits are limited from the behavior of the skin, due to high risk of dog ears.” This should be correlated with some references or demonstration by your own cases.

Reviewer 2 Report

Dear authors,

Some comments, though:
- the whole article needs to be extensively edited for the English language. There are a some spell errors and typing errors.

-References are not up-to-date (references between 1989 -2017). No actual literature is available that underlines the aim including the results /conclusion of this manuscript. Moreover, book references are used often (e.g. references 1 & 2). Topic-based references from publications are available and necessary. References are used twice (e.g. 4 and 6 Gottlieb et al).

- Figures (figures 1-4) are too small to read and quality is poor

Abstract:

- Abstract: No clear structure (Introduction, Methods, Results, Conclusion). Insufficient information regarding methods and results (e.g. which kind of skin lesions, how do the authors evaluated the patient acceptance, patient information etc.)

Introduction

- The introduction is insufficient. Reasons for this study or the gap this study wants to close are not clear. The whole study as well as the introduction refers to the reconstrutive ladder. There is a more actual concept. An extension of the concept of the reconstructive ladder and reconstructive elevator is the reconstructive clockwork. No literature is mentioned which supports the title / aim of this study. Hereof, scientific aspect / base is lacking in this study. A common thread is missing. Moreover, some references are missing (lines 30-33: after reference 1,2 --> reference 5,6 are following.

Materials and Methods:

- Materials and Methods are insufficiently reported. Why did the authors only use one question (likert scale 1-5) to evaluate the satisfaction? Why did the authors not used a validated questionnaire - e.g. for scar quality the POSAS (The Patient and Observer Scar Assessment Scale). This is a big methodical flaw in this study, thus, a conclusion only based on one question is insufficient. 

- Geometrical analysis and surgical techniques are poorly described. It is not reproducible. 

- Why did the authors include neoplasms of the limb and sternum (e.g. sternum is known to develop more hypertrophic scars)? Comparability is not given. 

- Explanation why ethical board approval was not necessary ("as per local policies on clinical studies"). The authors evaluated a new surgical technique on 15 patients; for the reader, it is not clear, why the authors didn't need informed consent from the patients (application of a new technique) or why ethical approval is necessary.

- No statistical analysis is described. For the reader it is not clear, how the analysis was done (table 1). More clarification is need; what does "higher is better" mean in this context (higher % means "parallelogram excision" is better, or vice versa? Use only one decimal digit. 

Results:

- Patients demographics are insufficient. No standard deviation is described. Only 15 patients were included. Here, the median and interquartile range are more suitable.

Discussion:

-  The discussed part in regard to the new technique is very scarce. The results are not discussed in this section. No limitations of the study and suggestions for future studies are described. No recommendation regarding the "new technique" is described (e.g. which patients can benefit, which patients do not). Moreover, no advantages and disadvantages and comparison between the "standard" methods are mentioned. 

Taken together, this manuscript has a lot of severe flaws in every section. The study design and the results are doubtful, thus, the conclusion in regard to the superior effect of the "new technique" on the outcome after 6 months post-surgery is problematic. Hence, I can't recommend this manuscript for publication.

Extensive editing of the English language is required.

Reviewer 3 Report

This is an interesting paper reporting the parallelogram excision. We believe that it is worthy of publication even as it is, but we recommend presenting representative photographs to make it easier for readers to understand.

Reviewer 4 Report

The authors are presenting a modification for the planning of surgical incisions for small excisions <=2cm which on most of the body are not a great surgical challenge.

The paper starts with a long introduction about the reconstructive ladder, however this is not a new rung, it is merely a modification of primary closure.

Comparison to flaps in not appropriate.

The diagrams were not clear in the manuscript I got to review.

There is a lack of data on the clinical cases, why were the age limits chosen, what were the indications and demographics of the patients?

Why are there no patient photos included.

The table with the mathematical data lacks information on the size ad configuration of the “lesions” furthermore there are no units reported/

I found the discussion and conclusion long and not supported by the scant clinical results.

None

Reviewer 5 Report

This is an interesting and inventive, though in its current form rather theoretical work about a novel excision technique for skin leasion aiming to simplify wound closure.

First of all, the manuscript is well written and the considerations behind the technique are presented in a comprehensible manner.

However, the manuscript lacks clinical pictures that support the considerations. A step-by-step instruction with clinical pictures would enhance the quality of the manuscript. The authors furthermore state that scar quality was good at last follow up in the majority of patients. Clinical follow up pictures would be of high interest to the readership.

Although obviously novel, the real benefit over a standard eliptical excision (that often includes correction of dog-ears, which in return may often result in a parallelogram-like shaped skin defect) is not entirely plausible. The whole discussion is based upon the consideration, that an eliptical excision automatically has to follow RSTLs, which is not always true. Consequently, also the amount of removed skin may not be much less with the parallelogram technique. This should be taken into consideration in the discussion section.

Compliments go to the authors for their work.

Round 2

Reviewer 1 Report

The authors did a through revision and my concerns have all been addressed. Congratulations to their great work!

Author Response

Dear Reviewer 1, many thanks for your support and for your very important suggestions, which definitely improved our manuscript.

Reviewer 2 Report

Dear authors,

after the revision, the manuscript significantly improved. Thank you for implementing my suggestions and comments. 

Minor editing of English language required.

Author Response

Dear Reviewer 2, many thanks for your support, encouragement and suggestions, which had a huge impact in terms of improvement of the manuscript. We have refined our English.